# Upcycling poly(succinates) with amines to N-substituted succinimides over succinimide anion-based ionic liquids

Fengtian Wu [1,2], Yuepeng Wang[1,3], Yanfei Zhao[1,3], Shaojuan Zeng[4], Zhenpeng Wang[1], Minhao Tang[1,3], Wei Zeng[1,3], Ying Wang[1,3], Xiaoqian Chang[1,3], Junfeng Xiang [1], Zongbo Xie[2], Buxing Han [1,3] & Zhimin Liu [1,3] ✉

The chemical transformation of waste polymers into value-added chemicals is of significance for circular economy and sustainable development. Herein, we report upcycling poly(succinates) (PSS) with amines into N-substituted succinimides over succinimide anion-based ionic liquids (ILs, e.g, 1,8-diazabicyclo[5.4.0]undec-7-ene succinimide, [HDBU][Suc]). Assisted with $H_2O$, [HDBU][Suc]) showed the best performance, which could achieve complete transformation of a series of PSS into succinimide derivatives and corresponding diols under mild and metal-free conditions. Mechanism investigation indicates that the cation-anion confined hydrogen-bonding interactions among IL, $H_2O$, ester group, and amino/amide groups, strengthens nucleophilicity of the N atoms in amino/amide groups, and improves electrophilicity of carbonyl C atom in ester group. The attack of the amino/amide N atom on carbonyl C of ester group results in cleavage of carbonyl C-O bond in polyester and formation of amide group. This strategy is also effective for aminolysis of poly(trimethylene glutarate) to glutarimides, and poly(1,4-butylene adipate) to caprolactone diimides.

As a kind of biodegradable plastics, poly(succinates) (PSS) produced from condensation coupling of succinic acid with diols, such as poly(ethylene succinate) (PES), poly(propylene succinate) (PPS), poly(butylene succinate) (PBS), poly(neopentyl glycol succinate) (PNGS) and poly(hexane succinate) (PHS), have been widely applied in our lives and industrial production due to their unique functions[1–3]. However, the accumulation of waste PSS as the result of low biodegradation in nature and emission of carbon dioxide has put serious impact on the environment. Therefore, the chemical degradation of PSS has been paid attention in recent years. For example, the lipase-catalyzed hydrolysis has been applied to degrade PSS-based polyesters

into succinic acid and corresponding diols[4–6], but suffering from complicated chemical processes and generation of oligomers. It is highly desirable to explore simple and efficient way to degrade spent PSS into value-added chemicals.

Succinimide and its N-substituted derivatives are important chemicals widely applied in organic chemistry, medicine, electroplate, and chemical analysis[7–9]. They can be produced via cyclizing reaction of succinic acid with ammonia and amines[10], and the reactions of succinimide with phenylboronic acid[11] or with halogenated hydrocarbon[12]. These approaches generally require expensive feedstocks, acid or base catalysts, and produce toxic byproducts, thus causing some problems

[1]Beijing National Laboratory for Molecular Sciences, Key Laboratory of Colloid and Interface and Thermodynamics, Center for Carbon Neutral Chemistry, Institute of Chemistry, Chinese Academy of Sciences, Zhongguancun North First Street 2, 100190 Beijing, P. R. China. [2]Jiangxi Province Key Laboratory of Polymer Micro/Nano Manufacturing and Devices, Jiangxi Province Key Laboratory of Synthetic Chemistry, East China University of Technology, Economic Development Zone, Guanglan Avenue 418, Nanchang 330013, P. R. China. [3]University of Chinese Academy of Sciences, Beijing 100049, P. R. China. [4]Beijing Key Laboratory of Ionic Liquids Clean Process, State Key Laboratory of Multiphase Complex Systems, Institute of Process Engineering, Chinese Academy of Sciences, Beijing 100190, P. R. China. ✉e-mail: liuzm@iccas.ac.cn

such as reactor corrosion and generation of wastes. Based on the chemical structures of PSS[13,14], upcycling PSS with ammonia or amines may be possible to access succinimide and its derivatives. However, no such report has been found yet in a literature survey.

As known as a kind of molten salts composed of organic cations and inorganic/organic anions, ionic liquids (ILs) are highly designable, and the multiple interactions (such as electrostatic interaction, van der Waals force, hydrogen bonding interaction and so on[15–19]) between the IL cation and anion cooperatively make ILs possess unique physiochemical properties. Therefore, ILs have been intensively applied as functional solvents and efficient catalysts in chemical reactions[20,21]. Especially, some task-specific ILs have been employed in dissolution and chemical degradation of waste polymers[22–24]. For example, acetate- and lactate-based ILs can dissolve polylactic acid and catalyze its hydrolysis, alcoholysis and aminolysis[22,23]. Choline-based IL is very effective for catalyzing polyethylene terephthalate degradation into chemicals[24].

In our previous work, it was found that the cooperation of the IL cation and anion as the hydrogen bond (HB) donor and acceptor, respectively, could achieve the cyclic dehydration of diols[25] and metathesis of aliphatic diethers[26], to access cyclic esters. In our continuous efforts dedicated to the ILs-catalyzed chemical reactions[27,28], herein we report a cation-anion confined hydrogen-bonding catalysis strategy to deconstruct PSS (e.g., PES, PPS, PBS, PHS) with amines into succinimide derivatives over succinimide anion-based ILs under metal-free and mild conditions (Fig. 1). Especially, assisted with small amount of water 1,8-diazabicyclo[5.4.0]undec-7-ene succinimide ([HDBU][Suc]) showed the best performance, which could achieve complete decomposition of a series of PSS (including PES, PPS, PBS, PHS) with amines, producing corresponding succinimide derivatives (e.g., 1-butylpyrrolidine-2,5-dione, 3a) and diols in high yields. The mechanism investigation indicates that the HB network formed among $H_2O$, the IL cation and anion, ester group, and amino or amide groups, strengthens the nucleophilicity of the N atoms in amino/amide groups, and improves the electrophilicity of carbonyl C atom in ester group. Confined by the electrostatic force between the IL cation and anion, the attack of the activated N atoms in amino or in the formed amide group on the activated carbonyl C atom of ester group results in cleavage of carbonyl C-O bond in ester group and formation of imide group. The cooperation of multiple hydrogen bonding interactions and electrostatic force between the IL cation and anion achieves the aminolysis of PSS into succinimide derivatives. This strategy is also effective for upcycling poly(trimethylene glutarate) (PTG) with amines to glutarimides, and (poly(1,4-butylene adipate) (PBA) to caprolactone diimides.

## Results

### Screening the IL catalyst and reaction conditions

To commence our investigation, the degradation of PBS with butylamine (2a) was selected as a model reaction to optimize the IL catalysts (Supplementary Fig. 1) and reaction conditions. As shown in Fig. 2, in the absence of any catalyst a little PBS was aminolysized, only producing N, N'-dibutylsuccinamide (4a) and 1,4-butanediol (BDO) in low

yields of 10%, while all the tested ILs were effective for catalyisng aminolysis of PBS with 2a, affording BDO in high yields, with amides 3a and 4a depending on the used ILs. The halide anion-based ILs including [EtMIm][Br], [EtMIm][Cl] and [COOH-EtMIm][Cl] (where [EtMIm] =1-ethyl-3-methylimidazolium, [COOH-EtMIm]=1-carboxyethyl-3-methylimidazolium) and 1-propylsulfonate-3-methylimidazolium trifluoromethanesulfonate ([SO_3H-PrMIm][OTf]) could afford 4a as the main amide product. Excitingly, the succinimide anion ([Suc])-based ILs including [P_{4444}][Suc], [N_{4444}][Suc], [Ch][Suc] and [HDBU][Suc] (where [P_{4444}]=tetrabutylphosphonium, [N_{4444}]=tetrabutylammonium, [Ch] =choline) gave 3a as the main imide product in high yields. In particular, [HDBU][Suc] provided the highest 3a yield of 80% (isolated yield: 78%). For comparison, sodium succinimide and potassium succinimide were examined for catalyzing upcycling PBS under the same other conditions, and no reaction occurred. This suggests that the cations of the effective ILs have contribution to the activity of the IL catalyst. The different performances of the tested ILs for PBS aminolysis may be ascribed to their unique chemical structures. Especially for the [Suc]-based ILs that can form dual hydrogen bonds (HBs) via the cation and anion as the HB donor and acceptor, respectively, they show different HB accepting behaviors though they have the same anion, reflected by the Kamlet-Taft parameter data (Supplementary Table 1). Considering that the electrostatic force between the IL cation and anion is a main interaction in the IL system, the electrostatic interaction and the HB accepting-donating behaviors of the ILs may jointly determine their catalytic activity.

After exploring the influences of temperature, reaction time, the amounts of used 2a, [HDBU][Suc] and water on the reaction (Supplementary Figs. 2 and 3), the optimized reaction conditions were obtained as follows: PBS (0.5 mmol structural unit), 2a (1.0 mmol), [HDBU][Suc] (0.5 mmol), $H_2O$ (1.0 mmol), 130 °C, 12 h. Notably, the 3a yields were remarkably influenced by the amounts of water, reaching the maximum value of 80% at the water amount of 1.0 mmol, i.e., the molar ratio of [HDBU][Suc] to $H_2O$ at 2:1, while dropping with further increase in $H_2O$ amounts (Supplementary Fig. 3). As known, $H_2O$ as HB acceptor and donor has strong ability to form HBs, which can form HBs with the cation and anion of [HDBU][Suc], thus mediating its catalytic activity. In the case of a small amount of $H_2O$, the H-bonding interaction between the IL and $H_2O$ can destroy the H-bonding interaction between the cation and anion, thus enhancing the activity of the IL. However, in the presence of a large amount of $H_2O$ the IL cation and anion may be surrounded by the $H_2O$ molecules, thus lowering its activity.

The thermal stability of the [Suc]-based ILs including [HDBU] [Suc], [P_{4444}][Suc], and [N_{4444}][Suc] was examined (Supplementary Fig. 4). It was indicated that the thermal stability of these ILs was significantly affected by their cations, among which [HDBU][Suc] showed the lowest decomposition temperature at 175 °C and [P_{4444}] [Suc] showed the highest one at 289 °C, identical to the reported results[29]. The recycling stability of [HDBU][Suc] for the degradation of PSS with aniline 2 h was explored, and it was demonstrated that [HDBU][Suc] almost remained unchanged activity after being reused for 4 times (Supplementary Fig. 5). This indicates that [HDBU][Suc] shows good stability in the depolymerization of PSS with amines.

### Aminolysis of PSS with various amines

Under the optimized conditions, various aliphatic and aromatic amines were employed in degradation of PBS over [HDBU][Suc] (Fig. 3a). It was indicated that most of the tested amines could achieve the complete decomposition of PBS, generating BDO in -100% yield (Supplementary Fig. 6) and succinimides in the yields of 64%-86%, accompanied with N, N-substituted succinamides as the byproducts (4a-4o and 4r, Supplementary Fig. 6). For example, using aqueous ammonia ($NH_3·H_2O$, 25 wt% in water) to degrade PBS, succinimide (3b) was obtained in a yield of 64%, which provides another route to

PES: n' = 1; PPS: n' = 2;
PBS: n' = 3; PHS: n' = 5;

PNGS

● simple, metal-free and efficient strategy
● mild conditions and wide substrate scope
● multiple hydrogen bonding interaction

**Fig. 1 | Our strategy for upcycling PSS.** Aminolysis of PSS into succinimide derivatives over [HDBU][Suc] with cation-anion confined hydrogen-bonding catalysis.

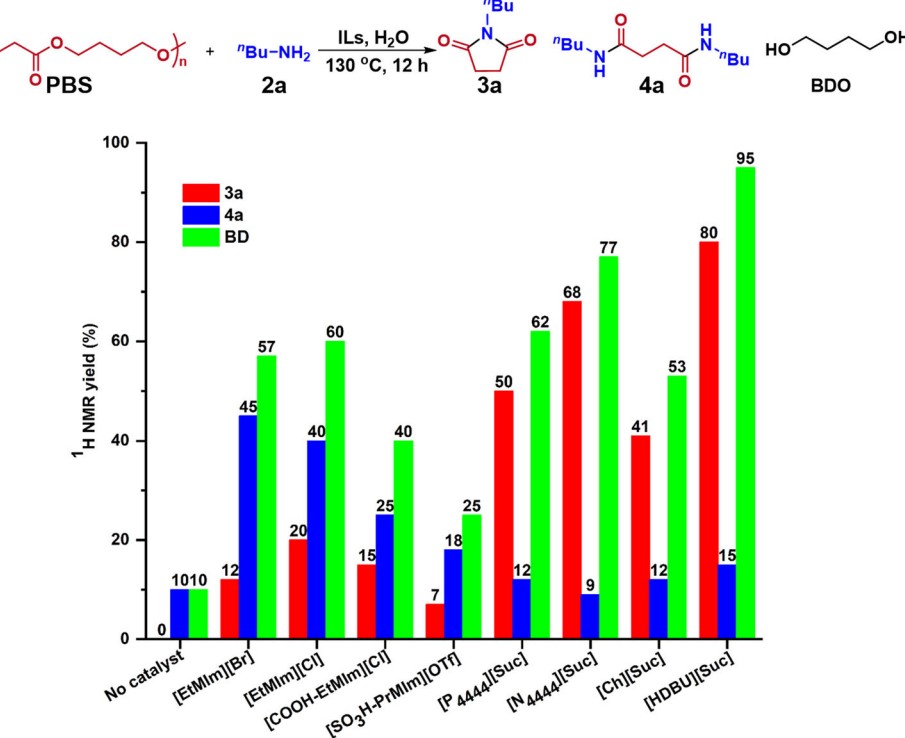

**Fig. 2 | Screening on IL catalysts using the aminolysis of PBS with 2a.** Reaction conditions: PBS (0.5 mmol structural unit), **2a** (1.0 mmol), ILs (0.5 mmol), $H_2O$ (1.0 mmol), 130 °C, 12 h; Product yields were determined by $^1H$ NMR using 1,3,5-trioxane as the internal standard, and the data were the average of three replicates with a reproducibility of ±3%.

synthesize this compound. Notably, the **3b** yield increased up to 72% in the case using [$P_{4444}$][Suc] as the catalyst. This suggests that the cations of the [Suc]-based IL catalysts could influence the selectivity of the catalyst towards succinimides. By comparison, aliphatic and aromatic amines showed comparable reactivity to degrade PBS and selectivity towards corresponding succinimides (**3a-3o** and **3r**, Fig. 3a). For the used aliphatic amines, their reactivity increased with the alkyl chain (**3a-3f**, Fig. 3a), and hexylamine (**2f**) afforded the highest yield (86%) of N-hexyl succinimide (**3f**). This trend may be related to the nucleophilicity of amino N in aliphatic amines, which reduces with increase in alkyl chain length of these aliphatic amines. For the tested anilines, the methyl-, methoxyl-, halogen-, and 4-thioanisole-substituted anilines were effectiv for degrading PBS (**3g-3o** and **3r**, Fig. 3A), affording the succinimides in yields around 70%, while 4-aminoacetophenone **2p** and p-nitroaniline **2q** showed no reactivity, prorably due to the strong electron-withdrawing effects.

In addition, the scale-up experiment for degrading PBS (10.0 mmol structural unit) with **2a** (20.0 mmol) was performed under the optimal conditions, and 1.16 g of **3a** was obtained in 75% yield, suggesting great potential for practical applications.

Similarly, the deconstruction of poly(trimethylene glutarate) (PTG) with amines under the catalysis of [HDBU][Suc] could produce N-aryl glutarimides in high yields (**5a-5o**, Fig. 3b), with 1,3-propanedi-carboxamides as the main byproducts (**6a-6l** and **6o**, Supplementary Fig. 7). Notably, these amines showed similar reactivity to degrade PTG compared to degrade PBS.

Besides PBS and PTG, other PSS including PES, PPS, PNGS, and PHS were also examined to be degraded by **2a** over [HDBU][Suc], which could be transformed into corresponding diols and succini-mides in high yields (Fig. 3c). The above findings indicates that upcy-cling PSS with amines over [HDBU][Suc] is an efficient way to decompose PSS, which also provides another route to access N-substituted succinimides.

What is more, we tried out the degradation of PTG and (poly(1,4-butylene adipate) (PBA) with $NH_3·H_2O$ (Fig. 3d), and the desired pro-ducts glutarimide **5a** and adipimide **7a** were obtained. In comparison, **5a** was accessed in a comparable yield (68%) to **3b**, while **7a** in a low yield (33%) with adipamide **8a** as the dominant product in a yield of 63%. These results suggest that the confinement effect between the IL cation and anion may play a key role in the formation of these cyclic imides. Comparing the structural units of PBA and PTG, the carbon chain length of PBA unit is shorter than that of PTG, which may be more matchable to the length of the IL cation and anion. This thus makes the cation-anion co-catalyzed cyclization reaction between two carbonyl carbon atoms of the structural unit with amino N occur effi-ciently, affording higher cyclic imide yield from PBA depolymerization.

**Reaction mechanism study**

In order to explore the reaction mechanism of PSS aminolysis to suc-cinimides, diethyl succinate (DS) was selected as a model molecule of the structural unit of PBS, and for convenience different amines including butylamine **2a**, hexylamine **2f** and aniline **2h** were used as amine models, to perform control experiments, in-situ FTIR and NMR analysis, and DFT calculations. In the control experiment of DS react-ing with **2a**, **3a** was obtained in a high yield of 85%, as expected (Fig. 4a). In order to determine the possible intermediates to form succinimides, the self-cyclization reactions of **4a**, 4-oxo-4-(phenyla-mino)butanoic acid (OPBA), and ethyl 4-oxo-4- (phenylamino) butanoate (EOPB), were tried, respectively, using [HDUB][Suc] as the catalyst. No reaction occurred for **4a** or OPBA, which excluded the possibility of **4a** or OPBA as the intermediates. The cyclization of EOPB took place, producing **3h** and ethanol in equivalent yields (Fig. 4b), which indicates that EOPB is the intermediate to form **3h**. Moreover, EOPB could react with **2h** to generate the corresponding succinamide **4h** and ethanol (Fig. 4c), which can explain the formation of N, N′-disubstituted succinamides in the aminolysis of PSS. From the these control experiments, it can be deduced that in the reaction process of

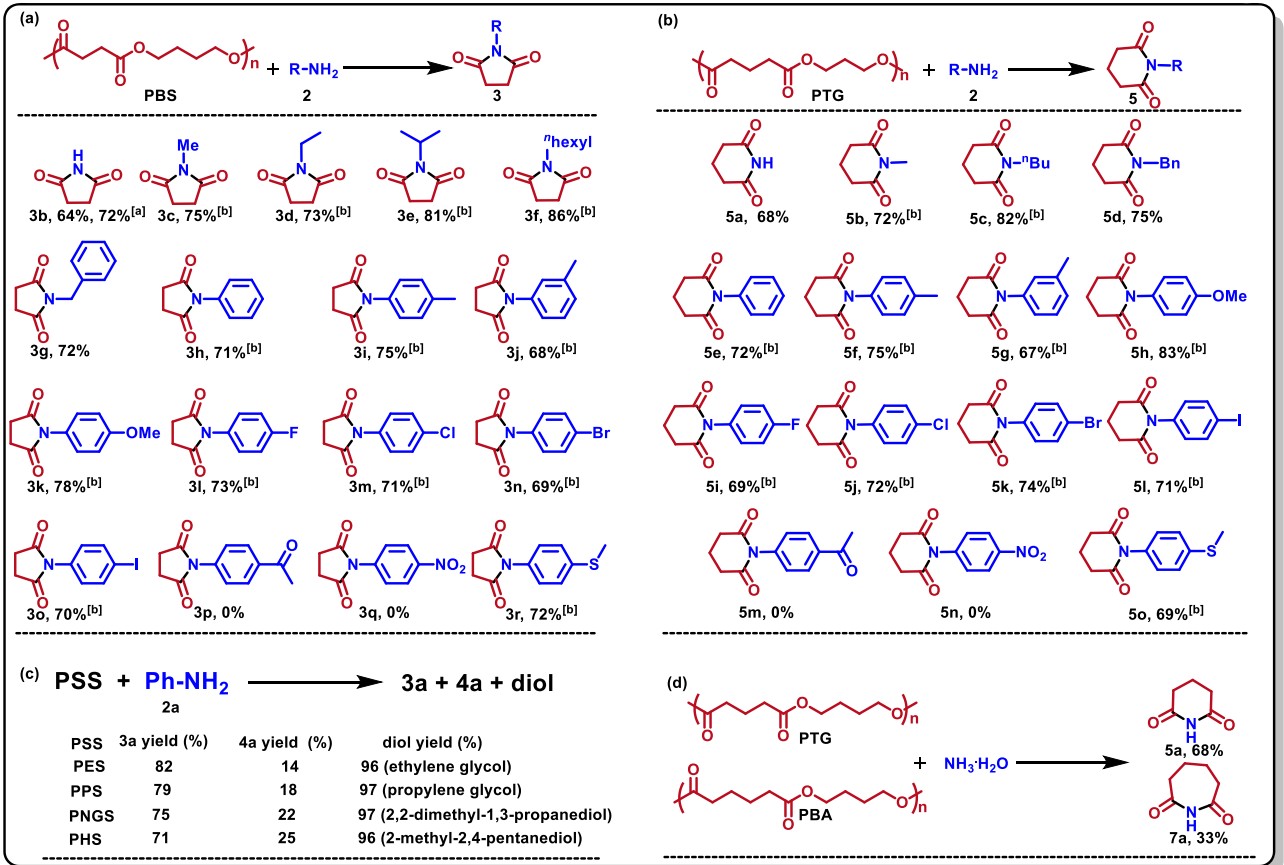

**Fig. 3 | Aminolysis of PSS with various amines over [HDBU][Suc]. a** Aminolysis of PBS; (**b**) Aminolysis of PTG; (**c**) Aminolysis of other PSS with **2a**; (**d**) Aminolysis of PTG and PBA with $NH_3 \cdot H_2O$. Reaction conditions: PSS (0.5 mmol structural unit), **2** (1.0 mmol), [HDBU][Suc] (0.5 mmol), $H_2O$ (1.0 mmol), 130 °C, 12 h. [a][P$_{4444}$][Suc] as the IL; [b]24 h.

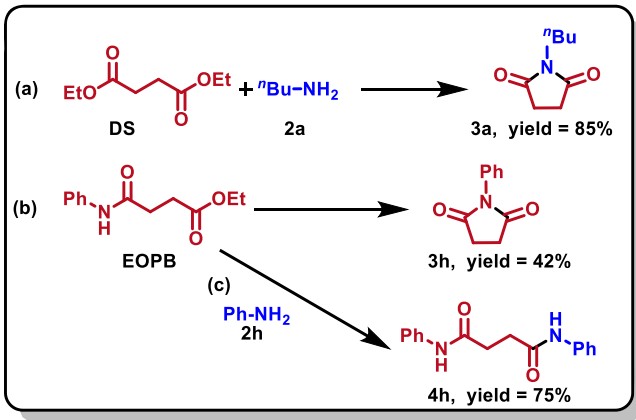

**Fig. 4 | Some control experiments. a** Reaction of DS with **2a** to **3a**; (**b**) Self-cyclization of EOPB to **3h**; (**c**) Reaction of EOPB with **2h** to **4h**. Reaction conditions: DS or EOPB (0.5 mmol), **2h** (1.0 mmol in e), [HDBU][Suc] (0.5 mmol), $H_2O$ (1.0 mmol), 130 °C, 12 h.

DS with **2a**, DS preferentially reacts with **2a** to form ethyl 4-(butylamino)-4-oxobutanoate intermediate, which further undergoes self-cyclization to form N-butyl succinimide, or reacts with **2a** to form byproduct **4a**. The control experiments are helpful to understand the aminolysis of PSS with amines.

In-situ FTIR and NMR spectroscopy analysis on DS reacting with **2 f** provides direct evidences on the degradation of DS and formation of the target product 1-hexylpyrrolidine-2,5-dione (**3f**) and the byproduct N, N'-dihexylsuccinamide (**4f**). As shown in Fig. 5a, the IR stretching vibration absorbance of the carbonyl group at 1735 cm$^{-1}$ assigning to DS decreased gradually as the reaction proceeded[30], meanwhile two peaks appeared at 1709 and 1680 cm$^{-1}$, which became more and more intense[31], ascribing to the IR stretching vibration absorbance of carbonyl groups in **3f** [32] and **4f**, respectively. In Fig. 5b, the altitudes of $^{13}$C NMR resonances at 171.92 ppm assigning to the carbonyl C atom and at 39.13 ppm to the alkyl C atom linking with N atom gradually enhanced with reaction time, meanwhile the altitude of $^{13}$C NMR resonances at 171.09 ppm assigning to the carbonyl C atom in DS decreased, indicating the degradation of DS and formation of **3f** and **4f**. In addition, the in-situ FTIR and NMR analysis on PBS aminolysis with **2f** were also performed, which offered similar results to the degradation of DS with **2f** (Supplementary Fig. 8).

To explore the roles of [HDUB][Suc] in aminolysis of PSS, the interactions of this IL with DS as one model of PSS structural unit, with **2f** or **2h** as amine models, with EOPB and OPBA as possible intermediates in the absence or presence of water, were investigated by means of NMR analysis and DFT calculations. From the $^{13}$C NMR spectra shown in Fig. 5c, it is obvious that the chemical shifts of carbonyl C atom in DS moves from 170.70 to 171.12 ppm as DS mixes with IL and further to 171. 27 ppm as $H_2O$ is present in the mixture. This indicates that the electron cloud density of the carbonyl C atom in DS declines and its electrophilicity improves, which is ascribed to the HB formation between [HDUB]$^+$ and the carbonyl O atom in DS (Supplementary Fig. 9a, b). From the $^{15}$N NMR spectra displayed in Fig. 5d, it is clear that the chemical shift of the N atom in **2 h** shifts from 56.80 to 55.75 ppm as **2 h** mixes with IL and further to 55.32

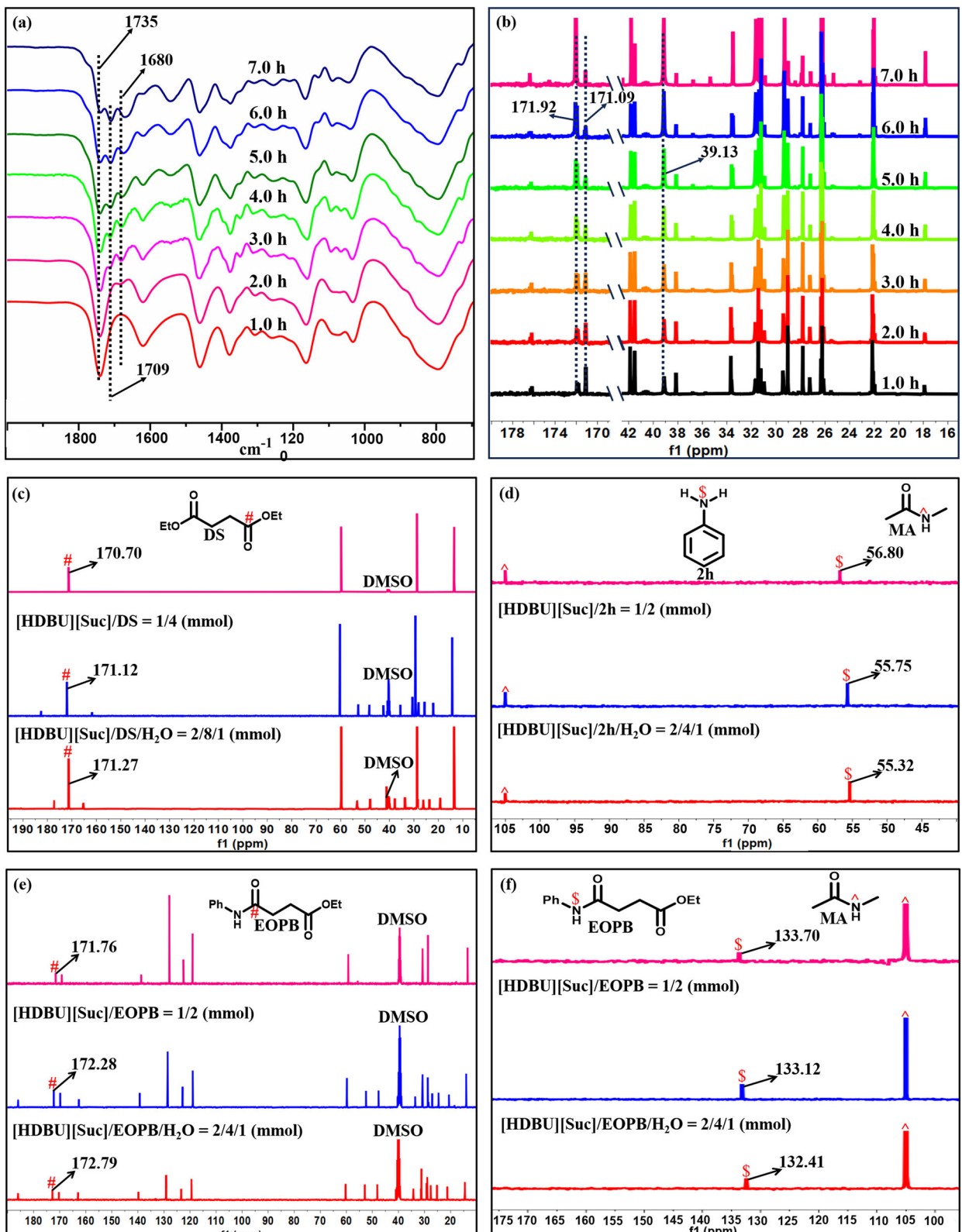

**Fig. 5 | Mechanism study.** (**a**) In-situ IR spectra recorded in the reaction of DS with **2f** performed at 120 °C; (**b**) in-situ $^{13}$C NMR spectra recorded in the reaction of DS with **2 f** performed at 120 °C; (**c**) $^{13}$C NMR spectra of DS, [HDBU][Suc]/DS, and [HDBU][Suc]/DS/H$_2$O recorded at 60 °C; (**d**) $^{15}$N NMR spectra of **2 h**, [HDBU][Suc]/ **2h**, [HDBU][Suc]/**2h**/H$_2$O recorded at 60 °C; (**e**) $^{13}$C NMR spectra of EOPB, [HDBU] [Suc]/EOPB, [HDBU][Suc]/EOPB/H$_2$O recorded at 60 °C; (**f**) $^{15}$N NMR spectra of EOPB, [HDBU][Suc]/EOPB, [HDBU][Suc]/EOPB/H$_2$O. Notes: 39.50 ppm of C in DMSO as the internal standard in **b**, **c** and **e**; 105.00 ppm for N in N-methylacetamide as the internal standard in **d** and **f**.

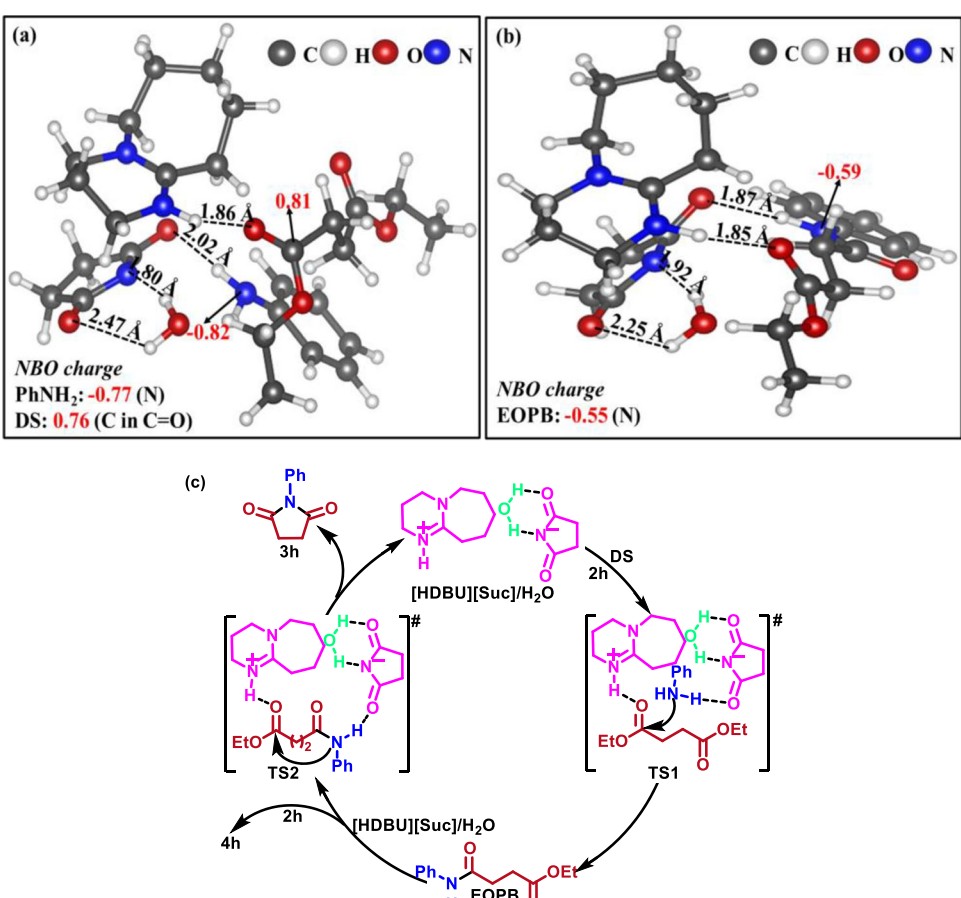

**Fig. 6 | DFT calculations for mechanism investigation and proposed reaction mechanism.** (**a**) Optimized geometries of [HDBU][Suc] interacting with H$_2$O, DS, and **2 h**; (**b**) Optimized geometries of [HDBU][Suc] interacting with H$_2$O, **2 h**, and EOPB; (**c**) The proposed reaction mechanism. Note: black font: atom distance; red font: NBO charges.

ppm as H$_2$O is added to the mixture. This indicates that the electron cloud density of amino N atom in **2 h** increases and its nucleophilicity improves, owing to the HB formation between the [Suc]$^-$ anion and the amino H atom in **2 h**, which is also supported by the $^1$H and $^{15}$N NMR analysis on the mixtures (Supplementary Fig. 9c and d). The $^1$H, $^{15}$N and $^{13}$C NMR analysis on the mixtures of EOPB/[HDUB][Suc] and EOPB/[HDUB][Suc]/H$_2$O provides evidences on the nucleophilicity enhancement of the amide N atom and the electrophilicity improvement of the carbonyl C atom in EOPB caused by the forma-tion of dual HBs of the IL cation with amide N atom and the IL anion with carbonyl O atom of EOPB (Fig. 5e, f and Supplementary Fig. 9e), respectively. The presence of the electrostatic force between the cation and anion makes it possible that the amide N atom in EOPB could attack the carbonyl O atom of ester group to achieve cycliza-tion, thus resulting in formation of succinimide compound **3 h**. For comparison, the $^1$H, $^{15}$N and $^{13}$C NMR analysis on the mixture of OPBA-[HDUB][Suc] was also performed (Supplementary Fig. 9f, g, h), which indicates the nucleophilicity of the amide N atom in OPBA is similar to that in EOPB activated by [HDUB][Suc]. Howerver, carbonyl C atom of ester in EOPB shows better electrophilicity than that of COOH group in OPBA, which may explain why [HDUB][Suc] could not catalyze the cyclization of OPBA to **3 h**.

To get more information on the interactions among the reaction species in the [HDUB][Suc]-catalyzed DS aminolysis with **2 h** in the presence of water, the DFT calculations were performed on a simple system that contains one molecule of each component. It is demon-strated that succinimide anion is preferable to form HB with H$_2$O ($\Delta H = -10.10$ kJ/mol), and the HB network among the IL cation, anion,

H$_2$O, DS, and **2 h** shows the lowest $\Delta H$ ($-31.94$ kJ/mol). From the opti-mized geometry (Fig. 6a), H$_2$O forms two HBs with succinimide anion: one between one H atom in H$_2$O and N atom in [Suc]$^-$ with HB length of 1.80 Å[23], and another between the other H atom in H$_2$O and O in [Suc]$^-$ with HB length of 2.47 Å[33] (Fig. 6a), which thus adjusts the interaction between [HDBU]$^+$ and [Suc]$^-$, probably generating favorable functions. Meanwhile, [Suc]$^-$ as the HB acceptor forms HB with one amino H atom of **2 h**, and [HDBU]$^+$ as the HB donor forms HB with one carbonyl O atom of DS. The length of HB formed between the carbonyl O atom of DS and amino H atom of cation was calculated to be 1.86 Å[23], which results in the NBO charge change of carbonyl C atom of DS from 0.76 for pure DS to 0.81 in the mixture (Fig. 6a), demonstrating the decreased electron cloud density and increased electrophilicity of the carbonyl C atom. The length of HB formed between the succinimide anion and one H atom in -NH$_2$ of **2 h** was 2.02 Å[34], and the NBO charge of N atom of **2h** changed from $-0.77$ for pure 2 h to $-0.82$ in the **2 h**/IL/DS/H$_2$O mixture (Fig. 6a), indicating the enhanced nucleophilicity of the N atom in **2 h**. In the cation and anion-confined microenvironment, the nucleophilic attack of the N atom in **2h** on the activated carbonyl C atom of one ester group may result in the cleavage of carbonyl C-O bond and formation of cyclic amide bond.

DFT calculations were also performed for the system composed of [HDUB][Suc], EOPB, and H$_2$O. Similarly, H$_2$O forms two strong HBs (length: 1.92 Å[35] and 2.25 Å[36]) with succinimide anion as marked in Fig. 6b. Meanwhile, two pairs of HBs are formed between [HDUB][Suc] and EOPB: one between amide H atom in EOPB and O atom in [Suc]$^-$ with length of 1.87 Å[23], and another between the H atom linking to N atom in [HDUB]$^+$ and carbonyl O atom in EOPB with length of 1.85 Å[23].

The dual hydrogen bonding interactions result in improvements of the nucleophilicity of the amide N atom and of the electrophilicity of the carbonyl C atom of EOPB, which is also demonstrated from the changes of the NBO charge of N atom of EOPB from −0.55 for EOPB to −0.59 for the EOPB/IL/$H_2O$ mixture. In the cation-anion confined electroneutral microenvironment, the nucleophilic attack of amide N atom on the carbonyl C atom of EOPB may occur to achieve cyclization of EOPB to form **3a**.

Based on the above results and discussion, a possible reaction pathway of DS reacting with **2 h** is proposed in Fig. 6c. Initially, with the assistance of water [HDUB][Suc] activates **2 h** and DS via hydrogen bonding interaction to form intermediate EOPB through transition state **TS1**. Then, EOPB is converted to target product **3 h** through intramolecular cyclization via transition state **TS2**, with release of [HDUB][Suc]/$H_2O$, meanwhile EOPB could react with **2 h** to generate byproduct **4 h**.

## Discussion

In summary, a simple, efficient, and metal-free protocol to upcycle PSS by amines over [HDBU][Suc] is developed, and a series of succinimides could be accessed in high yields under mild conditions (e.g., 130 °C). Mechanism investigation indicates that the presence of a small amount of $H_2O$ enhances the ability of [HDBU][Suc] to form HBs with ester, amino and amide groups, and thus catalyze the aminolysis of PSS in the cation and anion-confined electroneutral microenvironment. This strategy can also be extended to the aminolysis of PBA and PTG, producing corresponding diimides in high yields. The developed strategy not only can upcycle spent PSS, but also can provide another routes to access valuable diimide chemicals, which may have promising applications.

## Methods

### Materials and instrumentation

Poly(ethylene succinate) (power, Mw: 12000), poly(propylene succinate) (power, Mw: 9500), poly(butylene succinate) (power, Mw: 9000), poly(neopentyl glycol succinate) (power, Mw: 8000), poly(hexane succinate) (power, Mw: 1000), poly(trimethylene glutarate) (power, Mw: 2000) and (poly(1,4-butylene adipate) (power, Mw: 1000) were purchased from Shanghai Macklin Biochemical Co., Ltd. Ammonium hydroxide (25-28% in water), methylamine (40% in water), ethylamine (85% in water), isopropylamine (98%), butylamine (99%), hexylamine (99%) and benzylamine (99%) were provided by China National Medicines Co., Ltd. Aniline (99%), p-toluidine (99.5%), m-toluidine (98%), p-anisidine (99%), 4-fluoroaniline (99%), 4-chloroaniline (99%), 4-bromoaniline (99%), 4-iodoaniline (99%), 4-aminoacetophenone (99%), p-nitroaniline (99%), and 4-(methylmercapto)aniline (99%) were purchased from Beijing Innochem Science & Technology CO., Ltd. Diethyl succinate (98%) and 4-anilino-4-oxobutanoic acid were purchased from Shanghai Macklin Biochemical Co., Ltd. Silica gel (200-300 mesh) was provided by Qingdao Haiyang Chemical Co. Ltd. According to literature reports[37], ethyl 4-oxo-4-(phenylamino)butanoate (EOPB) was synthesized.

The ILs including [SO$_3$H-PMIm][OTf] (99%), [COOH-EtMIm][Cl] (99%), [EtMIm][Cl] (99%) and [EtMIm][Br] (99%) were provided by Centre of Green Chemistry and Catalysis, Lanzhou Institute of Chemical Physics (LICP), Chinese Academy of Sciences (CAS). Succinimide-based ILs including [N$_{4444}$][Suc], [P$_{4444}$][Suc], [Ch][Suc] and [HDBU][Suc] were synthesized via the neutralization of corresponding bases with succinimide[38]. The chemical structures of the as-synthesized succinimide ILs are shown in Supplementary Fig. 1. Sodium succinimide and potassium succinimide were purchased from Henan Alpha Chemical Co., Ltd.

Column chromatography was performed with silica gel (200-300 mesh) purchased from Qingdao Haiyang Chemical Co. Ltd. Thin-layer chromatography was carried out with Merck silica gel GF254 plates.

### Procedures for synthesis of ethyl 4-oxo-4-(phenylamino)butanoate (EOPB)

Ethyl 3-(chloroformyl)propionate (10.0 mmol, 1.64 g) and aniline (10.0 mmol, 0.93 g) were dissolved in 15 mL of dichloromethane loaded in a 50 mL glass tube reactor equipped with a magnetic stirring bar, and sealed with cap. After the solution was stirred at room temperature for 8 h, dichloromethane was removed via reduced distillation, and the residual solution was extracted with ethyl acetate (3×5 mL). The combined organic solutions were washed with water and brine, dried over anhydrous $Na_2SO_4$, followed by further drying in vacuo, affording the target product EOPB.

### Procedure for synthesis of [N$_{4444}$][Suc]

In a typical experiment to synthesize [N$_{4444}$][Suc], tetrabutylammonium hydroxide (10.0 mmol, 6.48 g), $H_2O$ (5 mL) and succinimide (10.0 mmol, 0.99 g) were mixed in a 50 mL glass tube reactor, equipped with a magnetic stirring bar and sealed with cap. Then, the mixture was stirred at room temperature for 24 h. After the reaction, the mixture was dried for removing most water in vacuo. Then, the residue was put in vacuum drying oven for 48 h at 60 °C, and [N$_{4444}$][Suc] was obtained. [P$_{4444}$][Suc], [Ch][Suc] and [HDBU][Suc] were synthesized following the analogous procedures.

### General procedure for aminolysis of PSS with amines

In a typical experiment to decompose PBS with aniline, PBS (86.0 mg, 0.5 mmol structural unit), $H_2O$ (1.0 mmol), and aniline (1.0 mmol) were mixed with [HDBU][Suc] (0.5 mol) in a 10 mL glass tube reactor, which was equipped with a magnetic stirring bar and sealed with cap. Then, the mixture was stirred at 130 °C for 12 h. After the reaction, the mixture was extracted with ethyl acetate (3 × 5 mL). The combined organic solutions were washed with water and brine, dried over anhydrous $Na_2SO_4$, and further dried in vacuo. The residue was purified by flash column chromatograph on silica gel to afford the target product **3h** and **4h**. The similar procedure is suitable for the synthesis of **3b-3o, 3r, 4a-4o, 4r, 5a-5l, 5o, 6a-6l, 6o, 7a**, and **8a**. Note: isolated yield of the product (wt%) = mass of the isolated product/theoretical mass of the product from complete conversion of PBS.

### NMR measurements

NMR spectra were recorded on Bruker Avance III 400 HD or 500 WB spectrometer equipped with 5 mm pulsed-field-gradient (PFG) probes. Chemical shifts are given in ppm relative to tetramethylsilane. To eliminate the effect of solvent, wilmad coaxial insert NMR tubes were used for [1]H, [13]C, [19]F, and [15]N NMR analysis at 393.2 K. DMSO-$d_6$ was added in the inner tube, and the sample was in the outer tube.

For NMR analysis of the products, the isolated products were dissolved in DMSO-$d_6$ and were analyzed on Bruker Avance III 400 HD or 500 WB.

For [1]H, [15]N NMR and [13]C NMR analysis, pure [HDBU][Suc], the mixture of [HDBU][Suc] and DS with a molar ratio of 1:4, the mixture of [HDBU][Suc] and **2 h** with a molar ratio of 1:2, the mixture of [HDBU][Suc] and EOPB with a molar ratio of 1:2, the mixture of [HDBU][Suc] and OPBA with a molar ratio of 1:2 were analyzed. Each sample (0.3 mL) was added into the outer tube, and the inner tube containing DMSO-$d_6$ was inserted. All NMR spectra were recorded on Bruker 500 WB at 333.2 K.

For in-situ [13]C NMR analysis, the mixtures of PBS (43.0 mg) or DC (0.5 mmol), 1-hexanamine **2 f** (0.5 mmol), $H_2O$ (0.5 mmol) and [HDBU][Suc] (0.25 mmol) were prepared. The sample (0.5 mL) was added into the NMR tube, and the spectra were recorded every 0.5 h from 0 to 7 h at 393.2 K. 7 or 4 representative in-situ [13]C NMR spectra were selected to analyze the intermediates during the reaction.

### In-situ infrared transmission characterization

The mixtures of PBS (43.0 mg) or DC (0.5 mmol), 1-hexanamine **2f** (10 mmol), $H_2O$ (5 mmol), and [HDBU][Suc] (0.25 mmol) were

prepared. FTIR spectra of the liquid samples were collected in the transmission mode on a Bruker Vertex 70 infrared spectrometer at a $1 \, cm^{-1}$ resolution, and the spectra were recorded every 0.5 h from 0 to 7 h at 393.2 K. 7 or 4 representative scans were collected for both samples.

### DFT calculations
All DFT calculations in this study were performed using Gaussian 16 package[39]. The structures of used compounds were optimized at the B3lyp-D3/6-31 g(d) level[40]. All calculated structures were verified with no imaginary frequency (IF). The molecular orbital information was calculated at the B3lyp-D3/def2-TZVP level. Thermal corrections were calculated within the harmonic potential approximation on optimized structures at 298.15 K and 1.0 atm. NBO analysis was performed to obtain the charge of each atom in every model molecule[41]. The VESTA molecular visualizing program was employed to draw 3D molecular structures. All 3D molecular views in this work are represented by ball and stick models, as white balls represent for hydrogen atoms, gray balls for carbon atoms, blue balls for nitrogen atoms and red balls for oxygen atoms.

**Note:** DFT calculation was performed to understand the interactions among [HDBU][Suc], $H_2O$, aniline and DS as a model compound of structural unit for PBS or EOPB as an intermediate. The DFT data presented have some limitations, which do not include great numbers of cations and anions to demonstrate the effect of cluster size, nonetheless, support the experimental results as they stand.

### General procedure for determination of Kamlet-Taft parameters for ILs[42,43]
For testing Kamlet-Taft polarity parameters of the succinimide-based ILs, $N,N$-diethyl-4-nitroaniline (DENA) was used to determine the value of $\pi^*$ that reflects the dipolarity/polarizability of ILs. Reichardt's dye (RD) and 4-nitroaniline (NA) were, respectively, employed as probes to determine the values of $\alpha$ and $\beta$, which reflect hydrogen bond acidity and hydrogen bond alkalinity, respectively. The concentrations of dyes DENA and NA in IL were $2.5 \times 10^{-5}$ and $1.0 \times 10^{-5}$ mol/L, respectively, and that of RD was $1.0 \sim 4.0 \times 10^{-4}$ mol/L. The adsorption spectra of the dye-IL mixture were recorded on a PE1050 UV–vis spectrophotometer. The Kamlet-Taft parameters were calculated using the equations: $\pi^* = 14.57 - 4270/\lambda_{max,DENA}$; $\beta = 11.134 - 3580/\lambda_{max,NA} - 1.125 \times \pi^*$; $\alpha = (19.9657 - 1.0241 \times \pi^* - \nu_{RD})/1.6078$; $\nu_{RD} = 1/(\lambda_{max,RD} \times 10^{-4})$, and the results are listed in Supplementary Table 1.

## Data availability
All data supporting the findings of this study are available within the paper and its Supplementary Information files. All data is available from the corresponding author upon request.

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

## Acknowledgements

The authors thank the National Natural Science Foundation of China (21890761, 22121002, 22233006) and Chinese Academy of Sciences (027GJHZ2022053MI) for their financial support.

## Author contributions

Z.L. directed the project and designed the experiments. F.W. performed the experiments. Y.W. performed the DFT calculations. S.Z. and F.W. determined the Kamlet-Taft parameters of ILs and analyzed the data. Y.Z., Z.W., M.T., W.Z., Y.W., X.C., J.X. and Z.X. analyzed the NMR and FTIR data. B.H. and Z.L. collaboratively discussed the reaction mechanism and contributed to the writing of the paper. All the authors participated in the discussion and preparation of the paper.

## Competing interests

The authors declare no competing interests.
