## [Peer Review File · Nature Communications]

Upcycling poly(succinates) with amines to N-substituted succinimides over succinimide anion-based ionic liquidsReviewers' Comments:

Reviewer #1:

Remarks to the Author:

In the manuscript "Upcycling poly(succinates) with amines to N-substituted succinimides over succinimide anion-based ionic liquids" by Wu et al., various polysuccinates were upcycled to nitrogen-substituted cyclic imides (succinimides) via aminolysis using several different amines and a succinimide-based IL as a catalyst. Through screening of several ILs and reacting conditions based on their ability to degrade poly(butylene succinate) (PBS) in the presence of butylamine, the authors found that 1,8-diazabicyclo[5.4.0]undec-7-ene succinimide ([HDBU][Suc]) along with H₂O at 130 °C for 12 h led to maximum yield of 1-butylpyrrolidine-2,5-dione, the desired product. The authors then investigated the effect of various aliphatic and aromatic amines on the degradation of PBS, poly(trimethylene glutarate) (PTG), and other polysuccinates using [HDBU][Suc] along with H₂O. The authors then conducted in-situ FTIR and ¹H, ¹⁵N, and ¹³C NMR analysis to explore the reaction mechanism of the aminolysis of these polysuccinates to succinimides using diethyl succinate, 4-oxo-4-(phenylamino)butanoic acid (OPBA), and ethyl 4-oxo-4-(phenylamino)butanoate (EOPB) as model molecules. Lastly, the authors performed DFT calculations to gain more information on the interactions among the reacting species in these IL-catalyzed aminolysis reactions. This manuscript does offer a unique avenue for the metal-catalyst-free upcycling of a variety of polysuccinates; however as detailed below, there are some issues with the provided analysis detailed in the manuscript. It is possible for this manuscript to be considered for publication in Nature Communications pending revisions – but one does note that the class of materials does not have the volume or relative importance of many other commercial polymers, so the manuscript's importance is somewhat muted.

Main Text

1. P. 3 (lines 96-97) – The authors state that the differences in PBS aminolysis with the varying ILs may be related to their chemical structures and their ability to form HBs. Can the authors elaborate further on this? This simple statement does not provide sufficient explanation.

- For example, while [HDBU][Suc] does, based on the Kamlet-Taft parameters, have the highest hydrogen-bond-donating behavior (highest α value of 1.61), it has the second to lowest hydrogen-bond-accepting behavior with a β value of 0.57. Based on the mechanism shown in Fig. 6, it appears that both the hydrogen-bond donating and accepting behaviors would be important for forming cyclic imides. Therefore, for example, it is interesting that [P4444][Suc], which has lower hydrogen-bond-donating behavior (α value of 1.47), but higher hydrogen-bond-accepting behavior (β value of 0.79) compared to [HDBU][Suc], had a 30% lower yield of 3a vs. [HDBU][Suc]. These data suggest that there is something other than the ability of the ILs to form hydrogen bonds that is affecting succinimide formation.

2. P. 3 (lines 100-102) – As stated by authors and shown in Fig. S3 the yield of 3a is maximized with the use of 1.0 mmol of H₂O (i.e., 2:1 molar ratio of [HDBU][Suc]: H₂O) but exhibits a notable decrease with either a lower or higher amount of H₂O. Can the authors provide some explanation to why this is the case? The manuscript as written has no explanation for this trend.

3. P. 5 (lines 146-148) – The authors seem to infer that the confinement effect between the IL cation and anion is the reason for lower formation of the desired cyclic imide from the depolymerization of PBA compared to PTG. Wouldn't any confinement effects from the IL be applicable in each of the cases (e.g., depolymerization of PBS and PTG)? If so, can the authors provide an explanation as to why the cyclic imide yield from PBA depolymerization is notably lower than PTG when their chemical structures only differ by one carbon between the carbonyl groups?

4. P. 7 (lines 214-216) – In comparing the ¹H, ¹⁵N, and ¹³C NMR of EOPB and OPBA, the authors stated that the nucleophilicity of the amide N atom activated by [HDUB][Suc] in OPBA is weaker than that of the [HDUB][Suc] activated amide N atom in EOPB. However, the signal of the OPBA IL-

activated N is located at a chemical shift of 132.8 ppm, while the EOPB IL activated N is located at 133.1 ppm. These signals are at quite similar chemical shifts. While the chemical shift of the EOPB N is slightly higher than that of the OPBA N, this difference does not seem significant enough to completely obstruct self-cyclization. Do the authors have a plausible reason for this occurrence?

5. General – Did the authors conduct any thermal stability testing of [HDBU][Suc]? One report of the thermal stability of [HDBU][Suc] shows evidence of thermal degradation of this IL occurring below 130 °C.¹ Did the authors test for evidence of IL decomposition post reaction? – see as an example: Liu, F. S.; Guo, J.; Zhao, P. H.; Jia, M. K.; Liu, M. S.; Gao, J., Novel succinimide-based ionic liquids as efficient and sustainable media for methanolysis of polycarbonate to recover bisphenol A (BPA) under mild conditions. *Polym Degrad Stabil* 2019, 169.

Minor comments

6. P. 2 (line 73) – The authors have “amide group”, but should it be “imide group”?

7. P. 2 (lines 73-74) – The authors state that the mechanism studies are partially a result of “fitted steric hinderance”. What is this statement referring to specifically? There was no clear mention of this specific topic in the manuscript or supporting information.

8. P. 4 (lines 120-121) – The authors stated that the reactivity of the aliphatic amines increased with increasing alkyl chain length. The authors should provide an explanation for this trend.

Supporting Information

9. P. 5 (Figures S7f and S7h) – The labels for both bottom spectra indicate mixtures of [HDBU][Suc]/2a. Shouldn't these both be [HDBU][Suc]/OPBA?

Reviewer #2:

Remarks to the Author:

Comments

The work from Liu and coworkers demonstrated a simple, metal-free and efficient strategy to upcycle poly(succinates) (PSS) with amines into N-substituted succinimides over succinimide anion-based ionic liquids via a cation-anion confined hydrogen-bonding catalysis. The as-developed catalytic approaches could be extended to the aminolysis of a series of PSS into succinimide derivatives and corresponding diols in high yields. The catalytic mechanism was detailly studied by in-situ FTIR, NMR and DFT calculations. This work provides an alternative way to achieve efficient plastic-to-value added chemicals transformation particularly ionic liquids with unique features and integrated catalytic capabilities have been harnessed. The following comments are provided for the authors' consideration:

1. The upcycling of poly(succinates) (PSS) with amines was highlighted, which could produce value added N-substituted succinimides. What is the benefit of this approach compared with other PSS upcycling approaches with or without amines in terms of product values, PSS transformation efficiency, catalytic systems, reaction conditions, energy input...? A summary or comprehensive comparison will help to highlight the current procedure.
2. ILs are deployed as the catalysts, which is attractive considering the structure diversity and stability compared to the traditional organic base derived catalysts. The thermal stability of the as-deployed ILs and the cycling stability are lack. In addition, as alcohols are co-produced, efficient separation approaches are required regarding the residual PSS, succinimides and alcohol product, as well as the IL catalysts. As water was required during the aminolysis procedure, how about the stability of ILs in the presence of water? Any hydrolysis?
3. Fundamental understanding on the relationship between cation/anion effects and catalytic efficiency is still required. If hydrogen bonding formation plays critical role during the catalytic procedure, is it

possible to quantify the H-bonding formation capability and correlate with the catalytic efficiency? It's challenging but will provide insights in the IL structure engineering.

4. Have the authors tried the upcycling of "true plastic wastes" by ILs? It will be very interesting to see real applications with only IL and water being involved.

5. Figure 2 should be re-organized and classified to clearly show the influence of diverse cations and anions.

6. There are some mistakes in the manuscript, which should be checked carefully. For example,

- "Fig. S1" was not found in the text.
- In supplementary information, "Figure" should be changed into "Fig." to keep identical to that in the formal text.
- "Fig. S2 and S3" should be changed to "Figs. S2 and S3"
- The caption of Fig. S2 should be modified.
- In Fig. 4, as for the caption, "4a or OPBA" did not appear in the control experiments, which should be deleted.
- The caption of Fig. 5 should be checked carefully.
- In Supporting Information, Figure 3 followed Figure S7, which is confused. The spectra in this Figure are the same as those in Fig. 5 of the text.

Reviewer #3:

Remarks to the Author:

This work describes upcycling poly(succinate) with amines by succinimide anion-based ionic liquids. The degradation of waste polymers such as poly(succinates) was important. Herein, the effect of ionic liquids, water, amines, polymers on the reactions was investigated. The results indicated protic ionic liquids with succinimide anion exhibited good performance. The mechanism was studied through the combination of spectroscopic investigation and DFT calculations. It can be considered for publication in Nature Communication after the following comments were addressed.

Firstly, this work shows dual-hydrogen bond play an important role in this reaction. Therefore, the literatures the role of multiple hydrogen bond in some catalytic reactions should be cited.

How about the separation and recycling of succinimide anion-based ionic liquids?

Secondly, how about the performance by succinimide anion-based salt, not ionic liquid? The comparison of this work with the literature was also important.

Response to Reviewers

Referee: 1

In the manuscript "Upcycling poly(succinates) with amines to N-substituted succinimides over succinimide anion-based ionic liquids" by Wu et al., various polysuccinates were upcycled to nitrogen-substituted cyclic imides (succinimides) via aminolysis using several different amines and a succinimide-based IL as a catalyst. Through screening of several ILs and reacting conditions based on their ability to degrade poly(butylene succinate) (PBS) in the presence of butylamine, the authors found that 1,8-diazabicyclo[5.4.0]undec-7-ene succinimide ([HDBU][Suc]) along with H₂O at 130 °C for 12 h led to maximum yield of 1-butylpyrrolidine-2,5-dione, the desired product. The authors then investigated the effect of various aliphatic and aromatic amines on the degradation of PBS, poly(trimethylene glutarate) (PTG), and other polysuccinates using [HDBU][Suc] along with H₂O. The authors then conducted in-situ FTIR and ¹H, ¹⁵N, and ¹³C NMR analysis to explore the reaction mechanism of the aminolysis of these polysuccinates to succinimides using diethyl succinate, 4-oxo-4-(phenylamino)butanoic acid (OPBA), and ethyl 4-oxo-4-(phenylamino)butanoate (EOPB) as model molecules. Lastly, the authors performed DFT calculations to gain more information on the interactions among the reacting species in these IL-catalyzed aminolysis reactions. This manuscript does offer a unique avenue for the metal-catalyst-free upcycling of a variety of polysuccinates; however as detailed below, there are some issues with the provided analysis detailed in the manuscript. It is possible for this manuscript to be considered for publication in Nature Communications pending revisions-but one does note that the class of materials does not have the volume or relative importance of many other commercial polymers, so the manuscript's importance is somewhat muted.

Comment 1: P. 3 (lines 96-97)-The authors state that the differences in PBS aminolysis with the varying ILs may be related to their chemical structures and their ability to form HBs. Can the authors elaborate further on this? This simple statement does not provide sufficient explanation. For example, while [HDBU][Suc] does, based on the Kamlet-Taft parameters, have the highest hydrogen-bond-donating behavior (highest α value of 1.61), it has the second to lowest hydrogen-bond-accepting behavior with a β value of 0.57. Based on the mechanism

shown in Fig. 6, it appears that both the hydrogen-bond donating and accepting behaviors would be important for forming cyclic imides. Therefore, for example, it is interesting that [P₄₄₄₄][Suc], which has lower hydrogen-bond-donating behavior (α value of 1.47), but higher hydrogen-bond-accepting behavior (β value of 0.79) compared to [HDBU][Suc], had a 30% lower yield of 3a vs. [HDBU][Suc]. These data suggest that there is something other than the ability of the ILs to form hydrogen bonds that is affecting succinimide formation.

Answer: We thank the reviewer very much for the valuable comment. Actually, it is difficult to answer the question what determines the activity of the ILs. As suggested, we reconsidered the roles of the IL catalyst in the aminolysis of PSS to succinimides. Besides the hydrogen-bond-donating or accepting behavior, the electrostatic force between the cation and anion that depends on the chemical structure of the ILs also plays important role in the reactions. From the Kamlet-Taft parameter data of the used ILs, it is clear that though [P₄₄₄₄][Suc] and [HDBU][Suc] have the same anion, their anions show different hydrogen-bond accepting behaviors due to the impact of the corresponding cations. This means the electrostatic force between the cation and anion influences the hydrogen-bonding accepting and donating behaviors of the ILs. Therefore, the cooperation of the dual hydrogen-bonding interaction and electrostatic force of the ILs determines their activities. In the revised manuscript, we addressed this.

Comment 2: P. 3 (lines 100-102) – As stated by authors and shown in Fig. S3 the yield of **3a** is maximized with the use of 1.0 mmol of H₂O (i.e., 2:1 molar ratio of [HDBU][Suc]: H₂O) but exhibits a notable decrease with either a lower or higher amount of H₂O. Can the authors provide some explanation to why this is the case? The manuscript as written has no explanation for this trend.

Answer: We thank the reviewer for the comment. Indeed, the use of H₂O significantly influences the reaction. As known, H₂O molecule as hydrogen-bond acceptor and donor can form strong hydrogen bond. The presence of H₂O in the ILs can mediate the ability of the ILs to form hydrogen bond. As for the used IL in this work, e.g., [HDBU][Suc], the hydrogen bonding interaction between the IL and H₂O is related to the amount of H₂O, thus showing the dependence of the activity of the IL on the amount of H₂O. In the case with small amount of H₂O, the H-bonding interaction

between the IL and H₂O can reduce the H-bonding interaction between the cation and anion, thus probably enhancing the activity of the IL. However, in the presence of large amount of H₂O the IL cation and anion may be surrounded by the H₂O molecules, thus lowering the activity of the IL. We reasonably discussed the role of H₂O in mediating the IL activity in the revised manuscript.

Comment 3: P. 5 (lines 146-148) – The authors seem to infer that the confinement effect between the IL cation and anion is the reason for lower formation of the desired cyclic imide from the depolymerization of PBA compared to PTG. Wouldn't any confinement effects from the IL be applicable in each of the cases (e.g., depolymerization of PBS and PTG)? If so, can the authors provide an explanation as to why the cyclic imide yield from PBA depolymerization is notably lower than PTG when their chemical structures only differ by one carbon between the carbonyl groups?

Answer: We thank the reviewer for the valuable comment. We infer that the confinement effect between the IL cation and anion is the main reason for the cyclic imide formation, which is applicable in depolymerization of both PBA and PTG. Comparing the structural units of PBA and PTG, the carbon chain length of PBA unit is shorter than that of PTG, which may be more matchable to the length of the IL cation and anion. This thus makes the cation-anion co-catalyzed cyclization reaction between two carbonyl carbon atoms of the structural unit with amino N occur efficiently, affording higher cyclic imide yield from PBA depolymerization. We tried to discuss this in the revised manuscript.

Comment 4: P. 7 (lines 214-216) – In comparing the ¹H, ¹⁵N, and ¹³C NMR of EOPB and OPBA, the authors stated that the nucleophilicity of the amide N atom activated by [HDUB][Suc] in OPBA is weaker than that of the [HDUB][Suc] activated amide N atom in EOPB. However, the signal of the OPBA IL-activated N is located at a chemical shift of 132.8 ppm, while the EOPB IL activated N is located at 133.1 ppm. These signals are at quite similar chemical shifts. While the chemical shift of the EOPB N is slightly higher than that of the OPBA N, this difference does not seem significant enough to completely obstruct self-cyclization. Do the authors have a plausible reason for this occurrence?

Answer: We thank the reviewer for the comment. The control experiments demonstrated that EOPB could achieve self-cyclization in the catalysis of [HDUB][Suc], while OPBA could not, from which it can be deduced that EOPB rather than OPBA was the possible intermediate to form succinimide. As for the reason why OPBA cannot self-cyclize to form succinimide, it is difficult to explain because it depends on multiple factors. The NMR analysis results indicate that the nucleophilicity of the amide N atom in **OPBA** is similar to that in **EOPB** activated by [HDUB][Suc], while the electrophilicity of the carbonyl C atom of the ester in **EOPB** is higher than that of the COOH group in **OPBA**. This may explain why [HDUB][Suc] could not catalyze the cyclization of **OPBA** to **3h**. In the revised manuscript, we discussed this.

Comment 5: General – Did the authors conduct any thermal stability testing of [HDBU][Suc]? One report of the thermal stability of [HBDU][Suc] shows evidence of thermal degradation of this IL occurring below 130 °C.¹ Did the authors test for evidence of IL decomposition post reaction? – see as an example: Liu, F. S.; Guo, J.; Zhao, P. H.; Jia, M. K.; Liu, M. S.; Gao, J., Novel succinimide-based ionic liquids as efficient and sustainable media for methanolysis of polycarbonate to recover bisphenol A (BPA) under mild conditions. *Polym Degrad Stabil* 2019, 169.

Answer: We thank the reviewer for the comment. As suggested, the thermal stability testing of the [Suc]-based ILs including [HDBU][Suc], [P₄₄₄₄][Suc], and [N₄₄₄₄][Suc] was performed. It was indicated that the thermal stability of these ILs was affected by their cations, among which [HDBU][Suc] showed the lowest decomposition temperature at 175 °C and [P₄₄₄₄][Suc] showed the highest one at 289 °C, as shown in Fig. S4. To explore the stability of [HDBU][Suc] for the degradation of PSS, [HDBU][Suc] was reused for 4 times in degradation of PBS with aniline **2h**, and it almost remained unchanged activity as illustrated in Fig. S5. This indicates that [HDBU][Suc] shows good stability in the depolymerization of PSS with amines under the experimental conditions though it displays a relatively low thermal stability.

Fig. S4. TGA curves of ILs: (a) [HDBU][Suc], (b) [N₄₄₄₄][Suc] and (c) [P₄₄₄₄][Suc]. Note: TGA analysis was performed under N₂ atmosphere with a flowrate of 20 mL/min and a heating rate of at 5 °C/min.

Fig. S5. The recycling experiments of [HDBU][Suc]. Note: reaction conditions: PBS (0.5 mmol structural unit), **2h** (1.0 mmol), [HDUB][Suc] (0.5 mmol), H₂O (1.0 mmol), 130 °C, 24 h; first time: 71%, second time: 68%, third time: 66%, fourth time: 69%.

Comment 6: P. 2 (line 73) – The authors have “amide group”, but should it be “imide group”?

Answer: We thank the reviewer for the careful review. In the revised manuscript, “amide group” was changed into “imide group” accordingly.

Comment 7: P. 2 (lines 73-74) – The authors state that the mechanism studies are partially a result of “fitted steric hinderance”. What is this statement referring to specifically? There was no clear mention of this specific topic in the manuscript or supporting information.

Answer: We thank the reviewer for the comment. We want to express that the length of the PBS structural unit is matchable to the length of the IL cation and anion. To avoid confusion, the phrase “fitted steric hinderance” was deleted in the revised manuscript.

Comment 8: P. 4 (lines 120-121) – The authors stated that the reactivity of the aliphatic amines increased with increasing alkyl chain length. The authors should provide an explanation for this trend.

Answer: We thank the reviewer for the comment. This trend may be related to the

nucleophilicity of amino N in aliphatic amines, which decreases with the increasing alkyl chain length of the aliphatic amines. This discussion was provided in the revised manuscript.

Comment 9: Supporting Information. P. 5 (Figures S7f and S7h) – The labels for both bottom spectra indicate mixtures of [HDBU][Suc]/2a. Shouldn't these both be [HDBU][Suc]/OPBA?

Answer: We thank the reviewer for the careful review. We have corrected it.

Referee: 2

The work from Liu and coworkers demonstrated a simple, metal-free and efficient strategy to upcycle poly(succinates) (PSS) with amines into N-substituted succinimides over succinimide anion-based ionic liquids via a cation-anion confined hydrogen-bonding catalysis. The as-developed catalytic approaches could be extended to the aminolysis of a series of PSS into succinimide derivatives and corresponding diols in high yields. The catalytic mechanism was detailedly studied by in-situ FTIR, NMR and DFT calculations. This work provides an alternative way to achieve efficient plastic-to-value added chemicals transformation particularly ionic liquids with unique features and integrated catalytic capabilities have been harnessed. The following comments are provided for the authors' consideration:

Comment 1: The upcycling of poly(succinates) (PSS) with amines was highlighted, which could produce value added N-substituted succinimides. What is the benefit of this approach compared with other PSS upcycling approaches with or without amines in terms of product values, PSS transformation efficiency, catalytic systems, reaction conditions, energy input...? A summary or comprehensive comparison will help to highlight the current procedure.

Answer: We thank the reviewer for valuable comments. Actually, this is the first work to upcycle PSS with amine into N-substituted succinimides. Compared to the commonly used hydrolysis approach to degrade PSS, which generally suffers from generation of oligomers and low conversion of PSS, this PSS upcycling approach with amines possesses obvious advantages such as complete conversion of PSS, production of highly valuable products, metal-free catalytic systems and mild conditions. In the

revised manuscript, we highlight the advantages of this approach in Conclusion section.

Comment 2: ILs are deployed as the catalysts, which is attractive considering the structure diversity and stability compared to the traditional organic base derived catalysts. The thermal stability of the as-deployed ILs and the cycling stability are lack. In addition, as alcohols are co-produced, efficient separation approaches are required regarding the residual PSS, succinimides and alcohol product, as well as the IL catalysts. As water was required during the aminolysis procedure, how about the stability of ILs in the presence of water? Any hydrolysis?

Answer: We thank the reviewer for valuable comments. As suggested, the thermal stability and recycling stability of the used ILs were examined. It was indicated that the thermal stability of these ILs ([HDBU][Suc], [N₄₄₄₄][Suc] and [P₄₄₄₄][Suc], Fig. S4) was related to their structures, among which [HDBU][Suc] showed the lowest decomposition temperature at 175 °C and [P₄₄₄₄][Suc] showed the highest one at 289 °C, as shown in Fig. S4. The recycling stability of [HDBU][Suc] was examined in degradation of PBS with aniline **2h**, and [HDBU][Suc] almost remained unchanged activity after used for 4 times as illustrated in Fig. S5. This indicates that [HDBU][Suc] shows good stability in the depolymerization of PSS with amines.

For the efficient depolymerization of PSS with amines, a small amount of H₂O was required, which enhanced the activity of the ILs due to the strong hydrogen bonding interaction between IL and H₂O. No hydrolysis products from PSS and from [HDBU][Suc] were detected in the reaction solution, probably ascribing to the presence of amines, which inhibited the hydrolysis reactions.

Regarding the separation of the resultant products, it is a bit difficult to separate. In this work, to separate the IL from the reaction solution, water was first added to dilute the reaction solution, and then ethyl acetate was added to extract organic products including generated alcohol and succinimides for three times. The recovered ethyl acetate solutions were combined and evaporated to remove ethyl acetate, and the mixture of alcohol and succinimide was obtained. Followed by separation via gel chromatography, alcohol and succinimide were separated. The collected aqueous phase was evaporated to remove water and dried at 65 °C for 24 h in vacuum, obtaining IL, which was used for the next run.

Comment 3: Fundamental understanding on the relationship between cation/anion effects and catalytic efficiency is still required. If hydrogen bonding formation plays critical role during the catalytic procedure, is it possible to quantify the H-bonding formation capability and correlate with the catalytic efficiency? It's challenging but will provide insights in the IL structure engineering.

Answer: We thank the reviewer for valuable comment. Indeed, it is challenging to quantify the H-bonding formation capability and correlated with the catalytic efficiency of the ILs, because the cooperation of electrostatic force and H-bonding capability of the IL cation and anion determines the activity of the ILs. We tried our best to provide information to reveal the relationship between the cation/anion effects and catalytic efficiency. However, due to the limitation of experimental conditions, we only performed in-situ NMR and FTIR analysis to explore the generated intermediate species and interactions among them, which could partially help to understand the relationship between the chemical structure and activity of the ILs.

Comment 4: Have the authors tried the upcycling of “true plastic wastes” by ILs? It will be very interesting to see real applications with only IL and water being involved.

Answer: We thank the reviewer for comment. As known, PBS plastics are generally used with other plastics, e.g. polylactic acid, to form plastics blend applied in package and tableware. Therefore, it is difficult to accurately degrade PBS used in package and tableware with our method, because of the influences from other plastics. In the revised manuscript, we carried out a scale-up experiment for degrading PBS (10.0 mmol structural unit) with **2a** (20.0 mmol) under the optimal conditions. The target product **3a** was obtained with 1.16 g in 75% yield, which showed great potential for practical application. This discussion was provided in the revised manuscript.

Comment 5: Figure 2 should be re-organized and classified to clearly show the influence of diverse cations and anions.

Answer: We thank the reviewer. As suggested, Figure 2 was re-organized and classified to clearly show the influence of diverse cations and anions.

Comment 6: There are some mistakes in the manuscript, which should be checked

carefully. For example:

- "Fig. S1" was not found in the text.
- In supplementary information, "Figure" should be changed into "Fig." to keep identical to that in the formal text.
- "Fig. S2 and S3" should be changed to "Figs. S2 and S3".
- The caption of Fig. S2 should be modified.
- In Fig. 4, as for the caption, "4a or OPBA" did not appear in the control experiments, which should be deleted.
- The caption of Fig. 5 should be checked carefully.
- In Supporting Information, Figure 3 followed Figure S7, which is confused. The spectra in this Figure are the same as those in Fig. 5 of the text.

Answer: We thank the reviewer for careful review. All the above errors mentioned by the reviewer were corrected in the revised manuscript. Moreover, the manuscript was checked and revised carefully to improve the English.

Referee: 3

This work describes upcycling poly(succinate) with amines by succinimide anion-based ionic liquids. The degradation of waste polymers such as poly(succinates) was important. Herein, the effect of ionic liquids, water, amines, polymers on the reactions was investigated. The results indicated protic ionic liquids with succinimide anion exhibited good performance. The mechanism was studied through the combination of spectroscopic investigation and DFT calculations. It can be considered for publication in Nature Communication after the following comments were addressed.

Comment 1: Firstly, this work shows dual-hydrogen bond play an important role in this reaction. Therefore, the literatures the role of multiple hydrogen bond in some catalytic reactions should be cited.

Answer: We thank the reviewer for the comment. As suggested, some relevant references on multiple hydrogen bonds were cited in the revised manuscript.

Comment 2: How about the separation and recycling of succinimide anion-based ionic liquids?

Answer: We thank the reviewer for the comment. In this work, to separate the IL from the reaction solution, water was first added to dilute the reaction solution, and then ethyl acetate was added to extract organic products including generated alcohol and succinimides for three times. The recovered ethyl acetate solutions were combined and evaporated to remove ethyl acetate, and the mixture of alcohol and succinimide was obtained. Followed by separation via gel chromatography, alcohol and succinimide were separated. The collected aqueous phase was evaporated to remove water and dried at 65 °C for 24 h in vacuum, obtaining IL, which was used for the next run. The separation procedures were provided in the revised supplementary information, as shown in the fig. S5.

Fig. S5. The recycling experiments of [HDBU][Suc]. Note: reaction conditions: PBS (0.5 mmol structural unit), **2h** (1.0 mmol), [HDUB][Suc] (0.5 mmol), H₂O (1.0 mmol), 130 °C, 24 h; first time: 71%, second time: 68%, third time: 66%, fourth time: 69%.

The recycling experiments showed that [HDUB][Suc] could be used for four times without activity loss.

Comment 3: Secondly, how about the performance by succinimide anion-based salt, not ionic liquid? The comparison of this work with the literature was also important.

Answer: We thank the reviewer for comment. As suggested, sodium succinimide and potassium succinimide were examined for the upcycling of PSB with aniline.

However, no desired product was detected. These comparison results were provided in the revised manuscript, which suggest that the IL cation also contributes to the activity of the ILs for catalyzing the decomposition of PSS with amines.

Reviewers' Comments:

Reviewer #2:

Remarks to the Author:

[Note from the Editor: Reviewer #2 was asked to assess also the response given to reviewer #1 who was not able to look over the revision again.]

The author addressed all my concerns and acceptance is recommended as the current status.

Most of the comments from Reviewer 1 is related to the structure-performance relationship of different ionic liquids being deployed, i.e., the relationship of hydrogen bonding formation capability of ILs with the catalytic activity, the stability of ILs during the reaction procedure. The authors do provide more details and reasonable explanation towards these questions. I have no more questions and recommend the acceptance of this paper at the current status.

Reviewer #3:

Remarks to the Author:

The authors have revised the manuscript carefully according to the reviewers' comments. Now it can be accepted.

Response to Reviewers

Reviewer #2 (Remarks to the Author):

[Note from the Editor: Reviewer #2 was asked to assess also the response given to reviewer #1 who was not able to look over the revision again.]

The author addressed all my concerns and acceptance is recommended as the current status.

Most of the comments from Reviewer 1 is related to the structure-performance relationship of different ionic liquids being deployed, i.e., the relationship of hydrogen bonding formation capability of ILs with the catalytic activity, the stability of ILs during the reaction procedure. The authors do provide more details and reasonable explanation towards these questions. I have no more questions and recommend the acceptance of this paper at the current status.

Answer: We thank the reviewer very much for the valuable comment.

Referee: 3

Reviewer #3 (Remarks to the Author):

The authors have revised the manuscript carefully according to the reviewers' comments. Now it can be accepted.

Answer: We thank the reviewer for the comment.